# Erythroid Cells as Full Participants in the Tumor Microenvironment

**DOI:** 10.3390/ijms242015141

**Published:** 2023-10-13

**Authors:** Julia A. Shevchenko, Kirill V. Nazarov, Alina A. Alshevskaya, Sergey V. Sennikov

**Affiliations:** 1Laboratory of Molecular Immunology, Federal State Budgetary Scientific Institution, Research Institute of Fundamental and Clinical Immunology, 630099 Novosibirsk, Russia; shevchenkoja2023@yandex.ru (J.A.S.); kirill.lacrimator@mail.ru (K.V.N.); 2Laboratory of Immune Engineering, Federal State Autonomous Educational Institution, Ministry of Health of the Russian Federation, Higher Education I.M. Sechenov First Moscow State Medical University, Sechenov University, 119048 Moscow, Russia; alshevskaya_a_a@staff.sechenov.ru

**Keywords:** erythroid cells, tumor immunosuppression, hypoxia, erythropoietin, transdifferentiation

## Abstract

The tumor microenvironment is an important factor that can determine the success or failure of antitumor therapy. Cells of hematopoietic origin are one of the most important mediators of the tumor–host interaction and, depending on the cell type and functional state, exert pro- or antitumor effects in the tumor microenvironment or in adjacent tissues. Erythroid cells can be full members of the tumor microenvironment and exhibit immunoregulatory properties. Tumor growth is accompanied by the need to obtain growth factors and oxygen, which stimulates the appearance of the foci of extramedullary erythropoiesis. Tumor cells create conditions to maintain the long-term proliferation and viability of erythroid cells. In turn, tumor erythroid cells have a number of mechanisms to suppress the antitumor immune response. This review considers current data on the existence of erythroid cells in the tumor microenvironment, formation of angiogenic clusters, and creation of optimal conditions for tumor growth. Despite being the most important life-support function of the body, erythroid cells support tumor growth and do not work against it. The study of various signaling mechanisms linking tumor growth with the mobilization of erythroid cells and the phenotypic and functional differences between erythroid cells of different origin allows us to identify potential targets for immunotherapy.

## 1. Introduction

The idea of using the capabilities of the immune system to treat oncological diseases dates back to the 19th century [1], and its further development led to the appearance of new therapeutic tools in clinical practice and an increase in patient survival [2]. However, despite the development of persistent remission in a large number of patients, there is a cohort of patients who still do not respond to treatment and eventually progress to the development of distant metastases [3]. The tumor microenvironment consists of many cell types that regulate tumor growth and progression [4]. The role of various subpopulations of T cells [5], myeloid suppressor cells [6], tumor-associated macrophages [7], neutrophils [8], and fibroblasts [9] in the regulation of antitumor immune responses was established after many years of research. It was shown that a population of erythroid progenitor cells with immunoregulatory properties could also be involved in the regulation of the local and systemic antitumor immune response [10]. Analyses of the heterogeneity of erythroid precursors from the yolk sac, fetal liver, umbilical cord blood, and bone marrow of humans showed that a part of the erythroid cell population had distinct immune properties and was present at all stages of human ontogenesis [11], while erythroid cells of different tissues significantly differed in the expression of genes associated with immunosuppression [12]. It is the immunomodulatory subset, rather than the entire population of erythroid precursors, that may represent the cellular basis of immunoregulatory functions [11]. In this review, the authors discuss the role of erythroid cells in the development and progression of cancer.

## 2. Erythropoietin and Metabolism Provide a Comfortable Existence for Tumor Cells

Cells of hematopoietic origin are among the most important mediators of the tumor–host interaction and, depending on the cell type and functional state, they exert pro- or antitumor effects in the tumor microenvironment or in adjacent tissues [13]. Both erythroid and myeloid hematopoietic cells take part in tumor microenvironment formation and cancer progression [14]. Erythropoietin(EPO), as the main mediator of erythropoiesis, is produced in tumor diseases not only by kidneys, but also by stromal tissue of the tumor, spleen, and liver to promote tumor growth by stimulating tumor angiogenesis and extramedullary erythropoiesis, resulting in increased oxygen perfusion [15]. Erythropoietin is the main regulator of definitive erythropoiesis, regulating the rate of erythropoiesis at rest and during stress. Erythropoietin acts through the receptor EpoR, a type I transmembrane cytokine receptor that first appears on the earliest erythroid-committed precursors and reaches a peak of expression with the beginning of terminal erythroid differentiation. EpoR protects proerythroblasts, basophilic erythroblasts, and CFUs (colony forming units) from apoptosis, which is the main mechanism of erythropoiesis rate regulation [16]. Erythropoietin and its receptor are expressed in various tumor tissues [17] and tumor cells [18]. The expression of erythropoietin receptor mRNA was shown in 90.1% of 65 melanoma cell lines, and an increase in the number of copies of EPO and EPO loci was observed in 30 and 24.6% of 130 primary melanomas, respectively. EPO knockdown in melanoma cells led to a decrease in EPO phosphorylation in response to EPO stimulation, a decrease in cell proliferation, and an increased response to the inhibitory effect of hypoxia and cisplatin in vitro [19]. Erythropoietin supports the survival of endothelial cells, neurons subjected to hypoxia; thus, it significantly supports the survival of tumor cells in breast cancer, head and neck tumors, and colorectal cancer. Due to its proliferative, antiapoptotic, and angiogenesis-stimulating activity, the use of erythropoietin in cancer patients is controversial [20]. Numerous studies have shown a negative effect of erythropoietin on the survival of patients with cancer [21]. For recombinant erythropoietin, which is used for the therapy of anemia, the presence of an alternative EphB4 receptor in tumor cells has been shown [22]. Recombinant erythropoietin, simultaneously with the action of ionizing radiation, led to the phenotypic conversion of non-stem cells into tumor-inducing breast cancer cells together with the re-expression of pluripotency factors c-Myc, Sox2, and Oct4 [23]. Knockdown of the erythropoietin receptor gene caused an increase in apoptotic activity in breast cancer cell lines and an increase in the expression of BimL and BimS, which are BH3 proteins of the Bcl-2 family only; these regulate the cell death program via the mitochondrial apoptotic pathway [24].

A study of the interaction between gliomas and erythropoietic organs showed that glioma cells remotely promoted erythroid differentiation in the bone marrow and spleen, including in the absence of anemia, through the production of humoral factors, even without direct contact. The conditioned environment of patient glioma cells promoted the maturation of TER119lowCD71high proerythroblasts into mature TER119highCD71low erythrocytes in vitro. The tumor leads to the induction of erythropoiesis and angiogenesis, which is the source of the red blood cell delivery of nutrients such as oxygen and iron to the glioma stem cell niche and promotes tumor progression [14]. Erythropoietin can interact with gliomas through stimulation of the antiapoptotic pathway or enhance stem cell proliferation through the STAT3 and Akt pathways [25,26,27]. The activation of the JAK2/STAT5, PI3K/AKT, and Ras/ERK pathways by erythropoietin promoted malignant cell behavior in the benign non-invasive rat cell line Rama 37 stably transfected by human EpoR [28].

Mature red blood cells use anaerobic glycolysis, in which one molecule of glucose is oxidized to form two molecules of ATP and two molecules of lactate. This pathway is the only source of metabolic energy for mature (anucleate) human erythrocytes. Most differentiated cells convert glucose to pyruvate through glycolysis and then to CO_2_ through the tricarboxylic acid cycle and oxidative phosphorylation in the mitochondria, with minimal lactate production. The transformation of a differentiated cell into a tumor cell leads to a metabolic switch: enhanced aerobic glycolysis (Warburg effect), in which glucose is catabolized to lactate under normoxic conditions. As a result of this glycolysis, large amounts of ATP are rapidly produced in a short time and mitochondrial activity is suppressed. Thus, the growing tumor cell proceeds to slow down the oxygen-dependent generation of ATP in the mitochondria and acquires a metabolism similar to that which has evolved in human erythrocytes, which lack mitochondria and other organelles. Such metabolic mechanisms allow tumor cells to maintain their proliferative autonomy, similar to mature erythrocytes [29].

## 3. Erythroid Cells Induce Immunosuppression during Tumor Growth

In a murine melanoma B 16 model, it was shown that environmental factors contributing to tumor development and hemodynamic changes caused by stress and tumors led to renal insufficiency, erythropoietin production, and the erythropoietin-dependent expansion of an immature spleen nucleated erythroid cell population. The cells with erythropoietin receptor expression were represented by a distinct fraction of cells dispersed throughout the intratumor space; these had nuclei, a TER119 marker, and a morphology similar to basophilic erythroblasts. No erythropoietin receptor expression was detected among the nucleus-free TER119+ cells. The transcriptome of tumor-induced erythroid cells significantly differs from resting bone marrow erythroid cells. Tumor-induced erythroid cells had a high expression of CD274 and other genes encoding immune checkpoints, Fam132b (also known as Erfe, or encoding erythroferrone), and Osm (encoding oncostatin M). The product of the CD274 PD-L1 gene was highly expressed on CD71+TER119+ tumor-infiltrating cells. A quantitative PCR analysis of purified CD71+TER119+ cells from B16 tumors showed a large number of Siglec3, CD300e, and CD244 transcripts, while the expression of erythropoietin receptors in these cells was comparable with that in bone marrow erythroid cells [30]. Melanoma growth leads to impaired erythropoiesis, severe anemia, and a five-fold increase in reticulocyte content, indicating an increase in erythrocyte precursor cells, which contrasts with a decrease in mature erythrocytes. A detailed analysis of the spleen and bone marrow of mice with melanoma showed an increase in the percentage of proerythroblasts and early erythroblasts as well as a decrease in late erythroblasts/reticulocytes compared with the spleen and bone marrow of normal mice [31].

The inoculation of Lewis carcinoma in mice resulted in an increase in spleen size due to CD71+TER119+ erythroid progenitor cells. Cells with the CD71+TER119+ phenotype constituted 5–10% of splenocytes in the mice without tumors and 40–60% of the total splenocytes in the mice with neglected tumors. During oncogenesis, erythroid cells also consistently accumulated in the bone marrow, liver, and blood, but not in lymph nodes or tumor tissues, and the number of erythroid cells in the spleen was higher than that of myeloid suppressors and T-regulatory cells by 3 and 30 times, respectively. Spleen erythroid cells with the CD71+TER119+ phenotype from mice on day 21 of the tumor growth were immunosuppressive; they suppressed T-cell proliferation by 50% at a suppressor:effector ratio of 1:1. A 2:1 ratio reliably suppressed CD8+ T-cell proliferation and inhibited CD4+ T-cell proliferation and Th1 differentiation [32].

A similar study with the grafting of hepatocellular carcinoma cells into mice showed that erythroid cells with TER119 expression were mainly present in the spleen, relatively absent in the tumor mass, and almost absent in the draining lymph nodes, bone marrow, liver, and lungs. In addition, the proportion of erythroid cells in the spleen increased with carcinoma progression and these erythroid cells appeared in the spleen of tumor carriers de novo rather than migrating from other sources. Thus, there was no induction of TER cells in organs, including tumor tissues, in Hepa-line mice with splenectomies and inoculated cells [32].

Smad is a family of structurally similar proteins that are the main signal transducers for the receptors of the transforming growth factor beta superfamily [33]. Smad3 is the most significantly phosphorylated signaling molecule in erythroid cells. Tumorigenic TGF-β is the main factor for the generation of erythroid tumor cells. An injection of neutralizing antibodies against TGF-β significantly inhibited the generation of TER spleen cells in mice with tumors, while the inoculation of Hepa cells with a TGF-β knockout induced far fewer erythroid spleen cells in vivo. Another factor that promotes tumor cell growth is the neurotrophic factor artemin, the expression of which is elevated in TER cells. A gene expression profile analysis showed that the genes of TER cells were significantly activated compared with lymphoid cells, especially in terms of artemin expression. Artemin induced the phosphorylation of a set of intracellular proteins and the most induced was the phosphorylation of caspase-9 at threonine position 125 (Thr125), which is crucial for inhibiting caspase-9 activation and initiating apoptosis, which is directly mediated by ERK activation [32] (Figure 1).

Erythroid cells contribute to tumor growth and progression. We present several mechanisms to realize this effect:Formation of the foci of extramedullary erythropoiesis in the spleen.Phosphorylation of proteins of the Smad family, which leads to activation of the expression of TGF-β, which has suppressor properties.Production of the immunosuppressive factor artemin.Suppression of the proliferation of CD8 T cells, suppression of proliferation, and Th1-differentiation of CD4 T cells.Activation of the genes of checkpoint molecules.

During erythroid development, the CD45 marker stops CD71+TER119+ cell precursors at undifferentiated stages. Its expression is a distinctive feature of early precursors [34]. The CD45−CD71+TER119+ population contains more terminally differentiated stage III–V erythroid cells, whereas CD45+CD71+TER119+ cells are more enriched in stage I–III precursors [35]. Early and late erythroid cells can have different expressions of genes related to immunosuppression, cell cycle progression, apoptosis, and glycolysis [12] (Figure 2).

In the spleen of mice with Lewis carcinoma, the ratio of CD45+CD71+TER119+ erythroid cells to CD4+ and CD8+ T cells was comparable with myeloid suppressors and much higher than for T-regulatory cells. Although CD45+ erythroid cells are present in anemia and in newborn mice, it is in tumors that they exhibit a more potent and robust suppressor activity. RNA sequencing (RNA-seq) showed that the global gene expression patterns of CD45+ erythroid cells were significantly different from their CD45 counterparts, especially for the signature genes that determine erythrocyte origins and immunosuppression. For the immunosuppression signature genes, myeloid suppressors and CD45+ erythroid cells were indistinguishable from myeloid suppressors. A subsequent analysis of these datasets revealed the enrichment of the reactive oxygen species pathway. The overproduction of reactive oxygen species is a well-established mechanism of myeloid suppressor-mediated T-cell immunosuppression, in which the ROS-generating NADPH oxidase family, including NOX2, plays an essential role [31].

In patients with hepatocellular carcinoma, the number of CD45+CD71+ erythroid cells in peripheral blood was significantly elevated compared with healthy subjects and patients with chronic hepatitis B (Figure 2). The CD45+CD71+erythroid cell population was the second most abundant population of suppressor cells after monocytic myeloid suppressors with the CD11b+HLA-DR−/lowCD14+CD15− phenotype. At the same time, there were far fewer erythroid cells with the CD45−CD71+CD235+ phenotype in peripheral blood compared with CD45+CD71+ erythroid cells. In hepatocellular carcinoma tissue, the content of CD45+CD71+ erythroid cells was almost 10 times higher than in peripheral blood. In para-tumor tissue at 2 cm from the focus, it slightly decreased; in normal liver tissue at 5 cm from the focus, it insignificantly differed from peripheral blood. Intratumor CD45+ CD71+ erythroid cells significantly reduced IFN-γ production and the proliferation of both CD4 and CD8+ T cells in a dose-dependent manner, whereas CD45–CD71+ CD235+ cells showed no suppressor activity. Erythroid cells with the CD45+CD71+ phenotype exhibit their immunosuppressive activity in a paracrine manner and through direct intercellular contact; their suppressor activity against T cells is significantly higher than that for myeloid suppressors [36]. The results of a complete transcriptome analysis by RNA sequencing showed the upregulation of a number of immunosuppressive molecules, including IL-10, TGF-β, and genes related to reactive oxygen species [37].

CD45+ circulating erythroid cells are the main cells expressing ROS and VISTA. The level of ROS expression in CD45+ circulating erythroid cells of a tumor microenvironment is significantly higher than in circulating peripheral blood erythroid cells and mature erythrocytes. At the same time, CD45+ circulating erythroid tumor cells have an altered membrane, which is manifested in their resistance to permeabilization [38].

A genome profiling analysis of single hepatoblastoma cells in children showed the foci of tumor-associated cells with a high proliferative activity and a high level of expression of erythroid cell genes, including HBB, HBG1/2, ALAS2, and erythroid precursors (KLF1). A gene signature ontology analysis showed that the erythroid signature was enriched for immune and detoxification processes. The identification of a hepatocarcinoma-associated population of erythroid cells suggests that in the postnatal liver, the tumor microenvironment could maintain a niche similar to the fetal liver, resulting in the preservation of erythroid progenitor cells. At the same time, cells from some patients expressed early erythroid markers and cells from other patients expressed late erythroid markers. Tumor-associated CD163+ macrophages expressing VCAM1 interacted with tumor-associated erythroid cells expressing ITGA4. IGF2-expressing tumor cells interact with IGF1R-expressing tumor-associated erythroid cells in the patient, and tumor cells expressing SPP1 interact with tumor erythroid cells expressing ITGA4. These data show that both tumor-associated macrophages and tumor cells interact with tumor-associated erythroid cells to maintain fetal erythropoiesis at different stages of development, supporting the hypothesis that the heterogeneity of hepatocarcinoma is partly associated with the fetal stage at which the tumor occurs [39].

Intratumoral erythroid cells of liver hepatoblastoma are more active in proliferation and differentiation and in the activation of pathways associated with metabolism compared with conventional erythroid cells, which are present in the liver tissue for several weeks after birth and hematopoiesis switching. Hepatoblastoma erythroid cells are similar to fetal liver cells in their high expression of liver-specific transcription factors (MYC, HDAC2, and HMGA1) and low expression of bone-marrow-specific erythroid genes (ZNF385D, CXCL8, KLF2, TNFRSF12A, and PRTN3). Thus, erythroid cells in hepatoblastoma simultaneously arise in the fetal liver with the development of the tumor and are not recruited from peripheral organs or bone marrow. The prolonged presence of erythroblasts in the liver during the development of hepatoblastoma is provided by the aberrant accumulation of erythroblast islands around VCAM1+ macrophages, which provide adhesion molecules and growth factors to promote the proliferation and survival of erythroid cells. One of the mechanisms of disruption of the antitumor T-cell immune response is to suppress the function of dendritic cells (DC) through the LGALS9/TIM3 axis. A blockade of TIM3 removes the inhibitory effect of erythroid cells on DC [40] (Figure 3).

The CD45 marker is normally expressed at early stages of the terminal differentiation of erythroid cells. In patients with CD45+ hepatocellular carcinoma/hepatoblastoma, erythroid cells are present in the tumor site, adjacent tissue, and peripheral blood.

Intratumor CD45+ erythroid cells are characterized by a high expression of fetal liver genes and a low expression of genes characteristic of bone marrow erythroid cells.CD45+ erythroid cells suppress T-cell proliferation both by contact and by the production of suppressive factors.The long-term survival and maintenance of the proliferative activity of CD45+ erythroid cells is provided by macrophages. In turn, CD45+ erythroid cells are able to suppress the development of dendritic cells through LGALS9/TIM3.During long-term tumor growth and metastasis, growth factor GM-CSF is able to induce the transdifferentiation of CD45+ erythroid cells into erythroid-derived myeloid cells (EDMC).

Anemia remains a major clinical problem in patients with cancer, leading to poor clinical outcomes and a reduced quality of life. Inflammation and proinflammatory cytokines such as tumor necrosis factor (TNF)-α, interleukin (IL)-6 and IL-1 contribute to the suppression of medullary erythropoiesis. Subsequently, anemia initiates extramedullary erythropoiesis as a compensatory mechanism, which leads to the expansion of circulating erythroid cells in the liver, spleen, and bloodstream. However, extramedullary erythropoiesis does not necessarily lead to the elimination of hypoxia, as the tumor manipulates erythropoiesis to convert erythroid progenitors into erythroid-differentiated myeloid cells in order to avoid antitumor immunity [38].

Mature extravascular red blood cells and hemoglobin also promote tumor growth. Tumor vessels are tortuous and irregular in shape with an uneven basement membrane and fewer covering pericytes, which contributes to tumor vessel leakage and hemorrhage. Red blood cells and hemoglobin stimulate tumor cell proliferation and the maintenance of the tumor stroma, which is characterized by a proinflammatory microenvironment enriched with M2 macrophages. In tumor tissue with the presence of red blood cells, levels of IL-6, IL-12, IL-1β, and TNF-α are increased. Endogenous TNF-α may act as a tumor promoter that activates the AP-1 and NF-κB signaling pathways, which stimulate cell proliferation and survival. Heme itself is an activator of TLR4. Red blood cells and hemoglobin can directly generate active IL-1β, which is required for angiogenesis and the invasiveness of various tumor cells in vivo by activating NF-κB to generate pro-IL-1β [41].

Mature red blood cells, under the influence of various factors, can undergo apoptosis-like suicidal cell death called eryptosis. The translocation of the cell membrane phospholipid phosphatidylserine to the cell surface, a decrease in cell volume, the formation of bubbles on the cell surface, and changes in membrane elasticity are characteristic signs of eryptosis. An increase in cytosolic Ca^2+^ concentration is the main mechanism of eryptosis, which results from the activation of Ca^2+^-permeable non-selective cation channels that mediate the influx of extracellular Ca^2+^ into erythrocytes [42]. In lung cancer patients, the level of phosphatidylserine-positive erythrocytes is two to three times higher than in healthy individuals [43]; inflammatory and metabolic remodeling in tumors enhanced the phagocytic activity of splenic phagocytes against phosphatidylserine-positive erythrocytes in tumor-bearing mice [44]. Erythropoietin, to compensate for anemia, can inhibit eryptosis, but high levels of erythropoietin promote the formation of red blood cells with a relatively high sensitivity to eryptosis triggers [45]. Eryptosis can be caused by oxidative stress. Reactive oxygen metabolites (ROS) such as singlet oxygen (^1^O_2_), superoxide anion (O^2–^), hydroxyl radical (^–^OH), and hydrogen peroxide (H_2_O_2_) [46] play an important role in the diversity of signaling pathways regulating cell migration, activation, and proliferation. These may be involved in the fight against viruses and pathogens, suppressing the proliferation of tumor cells. However, with excess production, the transfer of an unpaired electron leads to the oxidation of various molecules and cellular components as well as damage to proteins, lipids, and DNA [47]. Under the conditions of a dynamic imbalance between ROS and antioxidants, DNA damage typically occurs. The accumulation of DNA damage as a result of an incorrect or incomplete repair can lead to mutagenesis and, as a consequence, transformation, especially in combination with a deficiency in the apoptotic pathway [48]. Various sources of ROS during malignancies contribute to increased erythrocyte suicide and, therefore, anemia in cancer [49]. Circulating red blood cells are constantly exposed to reactive oxygen metabolites, so these cells have established a complex antioxidant defense system that includes non-enzymatic antioxidants such as glutathione and enzymatic antioxidants, including glutathione peroxidase, superoxide dismutase, and catalase [50]. However, red blood cells also contain NADFH oxidases, which can generate endogenous ROS, which leads to the oxidation of iron in hemoglobin (Fe^2+^) and conversion to methemoglobin (Fe^3+^) [51]. The autoxidation of hemoglobin leads to the formation of superoxide radical and hydrogen peroxide, which cause the formation of hemichromes, heme degradation, and the release of free iron, which catalyzes the formation of free radicals [52]. Methemoglobin is found in areas surrounding solid tumors. Under in vitro conditions, the release of hemin stimulates the proliferative activity of hepatocellular carcinoma and glioma lines. The process of hemoglobin oxidation in tumors during neovascularization and intratumoral hemorrhage can significantly contribute to an increased proliferation of tumor cells [53]. Anemia as a result of increased red blood cell turnover can lead to hypoxia [54], which causes tumor chemo- and radioresistance and also increases tumor aggressiveness and invasiveness [55] as well as ROS production, promoting tumor adaptation, survival, and persistence [56]. At the same time, antitumor therapy with cytotoxic drugs only enhances eryptosis. Most cytostatic drugs provide effective Ca^2+^ entry, increasing the ceramide content and causing oxidative stress and/or affecting kinase activity [57].

Tumors cause a systemic aberration in the myeloid lineage of development by directing the differentiation of erythroid progenitors into tumor-associated myeloid cells. A study of bone marrow cells from cancer patients showed that CD235a+ erythroid cells unexpectedly exhibited the transcriptome characteristics of myeloid cells. In bone marrow, peripheral blood, and pleura, myeloid surface markers (e.g., CD33 and CD11b) have not been shown to be actively expressed in CD235a+ erythroid cells. The relative size of this cell subpopulation steadily increases as the tumor progresses. This population can be identified as erythroid-modified myeloid cells (EDMC), with the phenotype CD45+/CD235a+/CD71+/CD11b+/CD33+/HLA-DR in cancer patients and CD45+/Ter119+/CD71+/CD11b+/Gr1+ in mice with tumors. The driving force behind this differentiation is believed to be GM-CSF produced by tumor cells, which can mediate the remote initiation of the transdifferentiation of erythroid cells into a myeloid lineage; intratumoral GM-CSF can create an inflammatory environment that facilitates the maturation of EDMC. The conversion of CD45+ circulating erythroid progenitors into EDMC may be one of the mechanisms by which extramedullary hematopoiesis does not replenish the pool of erythrocytes and does not compensate for hypoxia [58] (Figure 3).

Tumor-induced erythroid cells correlate with a poor tumor prognosis [31,32], so it has been suggested that tumor progression could be inhibited by targeting erythroid cells or their secreted products. The irradiation of Lewis carcinoma with local ionizing radiation led to a decrease in spleen size and a decrease in CD45– erythroid cells while maintaining CD45+ erythroid cells [59]. Exposure to ionizing radiation also led to a decrease in mRNA and artemin protein levels [59], which is an oncogenic factor associated with chemo- and radioresistance [60,61]. Local tumor irradiation does not decrease serum TGF-β concentrations, but stimulates local and systemic antitumor immunity through type I and II interferons (IFN) and CD8 T cells. The blockade of PD-1 creates a similar effect to irradiation [43]. Therapeutic approaches aimed at the surgical removal of the tumor, the spleen as a source of extramedullary erythropoiesis, the destruction of the tumor under the action of chemotherapeutic or immunotherapeutic drugs or radiation therapy lead to a decrease in the tumor load, which can significantly reduce the immunosuppressive effects of erythroid cells [62]. The powerful immunosuppressive properties of erythroid cells are temporary and disappear during differentiation. Consequently, the faster maturation of erythroid cells may be a favorable therapeutic factor to reduce immunosuppression [63].

## 4. Erythroid Cells Help Tumors Overcome Hypoxia

Tumor cells under hypoxia gain energy through glucose utilization and the stimulation of erythropoiesis and angiogenesis [64]. The tumor undergoes an initial period of avascular growth, followed by angiogenesis or vasculogenic mimicry, which binds to endothelium-dependent vessels to obtain sufficient blood and sufficient oxygen supply to ensure tumor growth, invasiveness, and metastasis [65,66,67]. Bone-marrow-derived erythrocytes are considered to be the main source of oxygen transport, but the ability to induce erythroid differentiation in vitro has been shown for human embryonic stem cells, inducing pluripotent stem cells using a combination of several transcription factors [68,69]. The uncontrolled overexpression of angiogenic factors in tumors leads to abnormal, loose, chaotically organized, immature, thin-walled, and poorly perfused vascular networks of the tumor, which leads to hypoxic conditions, even in highly vascular tumors [70].

However, the immunohistochemical detection of fetal hemoglobin in various solid tumors suggests an alternative origin for high-affinity hemoglobin [71]. In conditions of hypoxia and stress during genotoxic damage, cancer cells merge through endoreduplication and form giant cells with multiple copies of the genome [72]. A special population of tumor cells (PGCCs, or polyploid giant cancer cells) is formed, which is characterized by a large cytoplasm, aneuploidy, and multiple nuclei [73]. In various malignant tumors, PGCCs are a marker of enhanced stem cell properties, epithelial mesenchymal transition, and a more aggressive biology [74]. Polyploid giant cells in their development form a blastocyst-like structure similar to that observed during embryogenesis at the blastomere stage and acquire the ability to facilitate embryonic diapause, a reversible state of suspended embryonic development for survival in response to environmental stress. The exit from the embryonic diapause under the influence of therapeutic agents leads to the relapse or metastasis of the tumor [75].

Such erythroid cells of tumor origin can form structures of vascular mimicry during the development and progression of the tumor and express embryonic and fetal hemoglobin, which has a stronger affinity for oxygen than adult hemoglobin, which allows tumor cells to survive in hypoxia [76].

Several studies showed that the agent mimicking hypoxia, CoCl2, could induce the expression of stem cell genes characteristic of hematopoietic differentiation [77]. Fibroblasts and tumor cells treated with CoCl2 formed polyploid giant cells (PGCs) or polyploid giant cancer cells (PGCCs), which grew into spheroids with an activated expression of fetal hemoglobin, suggesting the presence of stem properties in these cells. In addition, different cell lines had different hemoglobin expression patterns. Ovarian fibroblast cells immortalized with hTERT and/or a p53 knockdown expressed only b/c/d/e hemoglobin. BT-549 breast cancer cells and human ovarian carcinoma cells expressed more embryonic and fetal hemoglobin. c-Myc is an oncogene of the four key pluripotency genes required for the production of induced pluripotent stem cells [78]. Immortalized cells overexpressing the c-Myc gene were positive for fetal and delta hemoglobin after CoCl2 treatment, indicating that c-Myc is a key regulator of cellular reprogramming toward the erythroid lineage [79]. CoCl2 can increase the expression level of c-Myc in immortalized cells by stabilizing the transcription induced by hypoxia factor 1α [80]. Under CoCl2-induced hypoxia, a population of cells with hematopoietic and erythroid markers CD34+, c-kit, and TER119 was detected among tumor stem cells [81]. A similar effect on the formation of polyploid giant cells and the formation of stromal—and, subsequently, erythroid—cells was demonstrated for the common chemotherapeutic drug paclitaxel [82]. In the tumor cells of colorectal cancer patients, it was shown that the expression of proteins associated with cell fusion (GCM1, syncytin-1, and ASCT-2) and the proteins associated with erythroid differentiation (hemoglobin-delta, hemoglobin-zeta, CD71, GATA-1, and GATA-2) increased from highly differentiated to low-grade and metastatic forms. Thus, the expression of proteins associated with cell fusion and the proteins associated with the differentiation of erythroid cells may be significant clinical signs in the prognosis of metastasis for tumors prone to the formation of polyploid giant cells [83].

In some tumor tissue samples of hepatocellular carcinoma, clusters of cells with multiple erythrocyte markers without an endothelial cell lining and vessel formation were found without the presence of blood vessel markers. The generation of such cells may be induced by tissue hypoxia, which is characteristic of many cancers, as shown by the activation of the transcription factor HIF1a in certain tissue regions of hepatocellular carcinoma. Hypoxia is further confirmed by the presence of the erythropoietin receptor in hepatocellular carcinoma tissue because erythropoietin, a key hematopoietic cytokine induced by hypoxia, is only temporarily expressed, but it controls erythropoiesis and protects cells from hypoxic damage [84]. The presence of clusters of erythrocyte-like cells at different stages of differentiation without any signs of surrounding blood vessels or endothelial cell linings suggests the transdifferentiation of hepatocytic carcinoma stem cells into erythrocytes. This is evidenced by the appearance of glycophorin A in epithelial cells resembling hepatocytic strands before adopting a mesenchymal state and the activation of cell signaling pathways related to erythropoiesis as well as the co-expression of cytokeratin 18 and glycophorin A. Most tumor tissues have stem cells that support cancer growth and persist after pharmacotherapy. Such cells show an expression of the transcription factors NANOG and OCT4, which regulate self-renewal and pluripotency with a transition to erythropoiesis and/or angiogenesis. The activation of NANOG, especially OCT4, in some cells in the regions adjacent to erythrocyte clusters and the co-expression of glycophorin A in the same cell further confirms the transdifferentiation of stem cells into erythrocyte cells, followed by enucleation and the death of such cells. The Wnt pathway of signal transduction and the activation of Sulf1/Sulf2 enzymes, which are usually not found in normal livers, is the main pathway for hepatocellular carcinoma cells [85]. The generation of erythroid cells from fibroblasts and tumor cells provides a rational explanation of how normal and cancer cells can obtain oxygen under hypoxia with or without angiogenesis. Thus, the ineffectiveness of conventional antiangiogenic cancer therapies targeting only endothelium-dependent vessels can be explained as tumor cells have the ability to directly generate erythroid cells and adapt to a hypoxic microenvironment. In addition, obtaining erythroid cells from immortalized cell lines can provide an alternative unrestricted source for in vitro erythrocyte production [86] (Figure 4).

## 5. Conclusions

In the pathogenesis of many types of cancer, erythroid cells have an immunosuppressive effect on immunocompetent cells of lymphoid origin and contribute to tumor growth and resistance to existing treatment methods. Despite the fact that erythroid cells have the most important function of life support in the body, they support the growth of the tumor and do not fight it. The presence and high content of erythroid cells in some types of human tumors can serve as a reliable marker for the prognosis of tumor recurrence. The study of various signaling mechanisms linking tumor growth to erythroid cell mobilization as well as the phenotypic and functional differences between tumor and bone marrow erythroid cells and their immunomodulatory properties can reveal potential targets for immunotherapy. The functional features of the interaction of erythroid and tumor cells lead to the need for new approaches in the use of erythropoietin for the treatment of anemia and the reasons for the ineffectiveness of antiangiogenic drugs. Exposure to erythroid cells or their effector mechanisms in combination with other treatment methods can significantly increase their therapeutic effectiveness. The study of various signaling mechanisms linking tumor growth with the mobilization of erythroid cells as well as the phenotypic and functional differences between erythroid cells of different origins allows us to identify potential targets for immunotherapy.

## Figures and Tables

**Figure 1 ijms-24-15141-f001:**
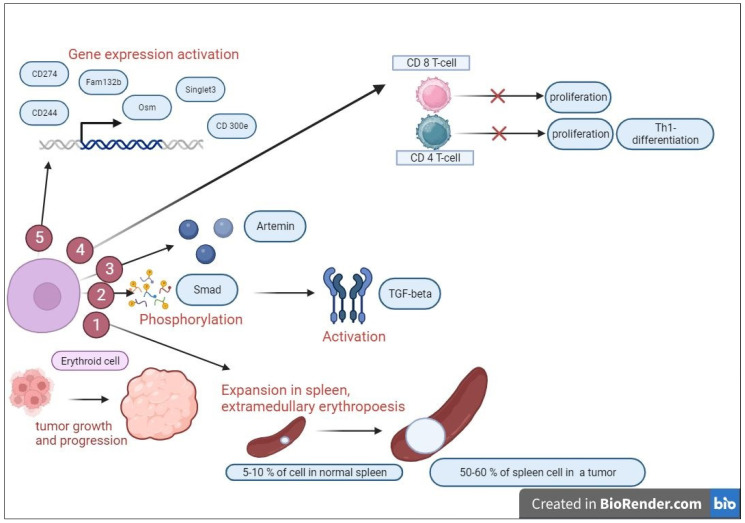
The main mechanism involved in maintaining tumor growth by erythroid cells.

**Figure 2 ijms-24-15141-f002:**
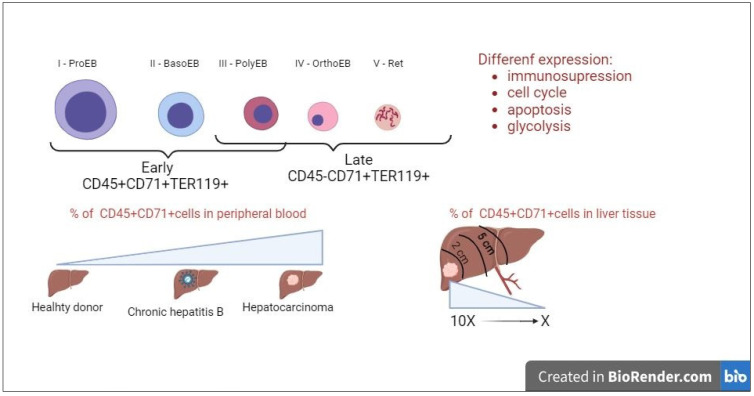
CD45+ erythroid cells are the main population of intratumoral erythroid cells.

**Figure 3 ijms-24-15141-f003:**
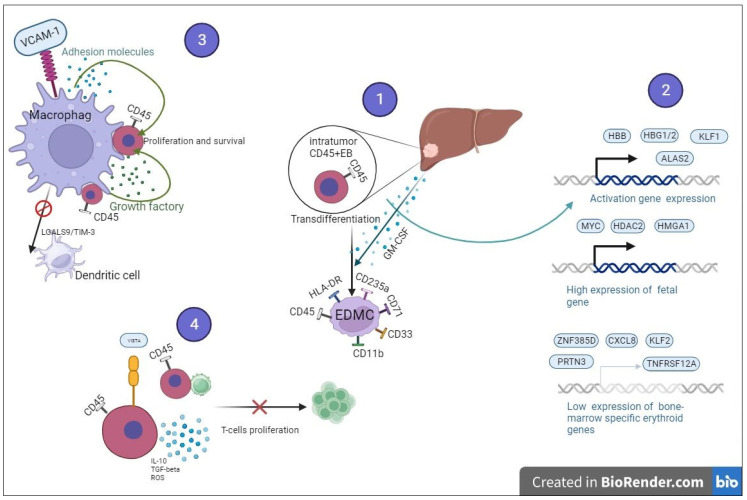
The main mechanism involved in maintaining tumor growth by CD45+ erythroid cells.

**Figure 4 ijms-24-15141-f004:**
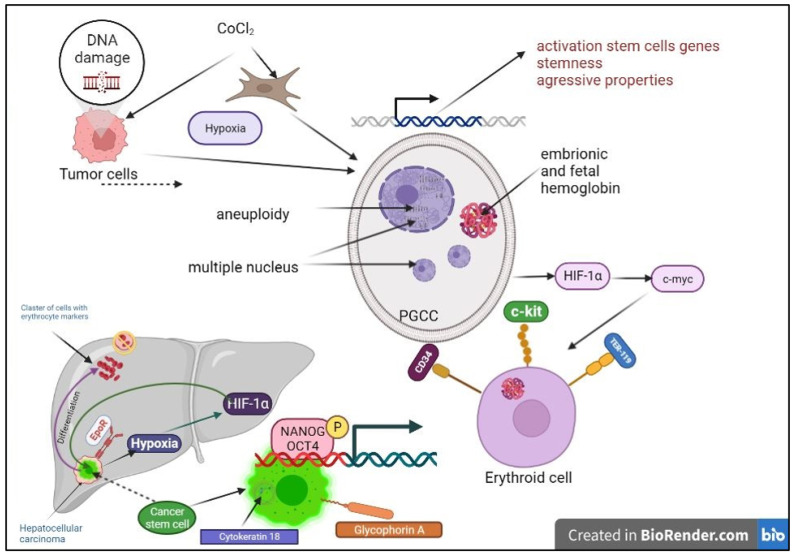
Erythroid cells help tumors to overcome hypoxia. Under conditions of hypoxia or damage by genotoxic agents, a population of polyploid giant cancer cells (PGCCs) appears in the tumor, which have stem properties and are able to differentiate into erythroid cells and form clusters of genes with erythroid markers in the absence of blood vessels.

## Data Availability

Not applicable.

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
