# Peer review of "Erythroid Cells as Full Participants in the Tumor Microenvironment"

_ijms, 2023, doi:10.3390/ijms242015141_

Round 1

Reviewer 1 Report

Shevchenko et al, reviewed the literature about the crosstalk between tumors and erythroid cells.

The manuscript is well conducted and organized, and could be of potential great interest to the scientific community.

It would be great if authors could answer some concerns before the final acceptance.

References 17 and 18 are crucial for this review because they reported the overexpression of erythropoietin and its receptor in tumor tissues and cells, supporting the crosstalk between tumors and erythroid cells. However, it’s really necessary to cite more references for different teams. Moreover, reference 17 is a bioinformatic analysis of different databases and it should be interesting to also add some references of experimental papers.

Line 82: authors have written “They deliberately avoid or turn off their respiration…”. In my opinion, this phrase should be rewritten to avoid any idea of “cell intentionality”

Figure 1 needs more detailed legend. Furthermore, figure title should be changed in “The main mechanism involved in maintaining tumor growth by erythroid cells”

At different place, they are wrong hyphenations (lines 112, 124, 142, 190, 211, 266)

Author Response

Shevchenko et al, reviewed the literature about the crosstalk between tumors and erythroid cells.

The manuscript is well conducted and organized, and could be of potential great interest to the scientific community.

It would be great if authors could answer some concerns before the final acceptance.

References 17 and 18 are crucial for this review because they reported the overexpression of erythropoietin and its receptor in tumor tissues and cells, supporting the crosstalk between tumors and erythroid cells. However, it’s really necessary to cite more references for different teams. Moreover, reference 17 is a bioinformatic analysis of different databases and it should be interesting to also add some references of experimental papers.

We have added more information, including experimental work, to this section. We have marked these fragments in green.

Line 82: authors have written “They deliberately avoid or turn off their respiration…”. In my opinion, this phrase should be rewritten to avoid any idea of “cell intentionality”

I have changed this sentence and made it more precise and clear. We have marked these fragments in green.

Figure 1 needs more detailed legend. Furthermore, figure title should be changed in “The main mechanism involved in maintaining tumor growth by erythroid cells”

We have changed the name of the drawing. We also really like this option. We have expanded the legend and added several clarifying fragments to the drawing itself. We have marked these fragments in green.

At different place, they are wrong hyphenations (lines 112, 124, 142, 190, 211, 266)

I have made the necessary corrections. I assume that such errors arose while working on multiple devices and editing text. Thank you for your attentiveness. We have marked these fragments in green.

Reviewer 2 Report

In their review titled " Erythroid cells as full participants in the tumor microenvironment" Shevchenko et al discuss the role of erythroid progenitors in the tumor microenvironment. This subject has been in the spotlight of cancer research in the last years, since erythroid progenitors seem to function in an immune-modulatory way in the tumor microenvironment, helping the cancer cells survive. Therefore, the topic the authors chose is original and relevant to the field of cancer biology and cellular and molecular sciences. While there are some recent reviews on this topic (maybe the authors could acknowledge them in their revised version, e.g., 10.1016/j.canlet.2023.216193, 10.1097/MOH.0000000000000762 and 10.1515/biol-2022-0102), the authors draw a quite complete picture of cancer cases in which the role of erythroid progenitors has been explored, including melanoma, hepatic carcinoma, and Lewis carcinoma, and discuss the pleiotropic ways in which erythroid cells act immunosuppressively (graphically shown in Figure 1). The authors also discuss the impressive yet monstrous ability of some cancer cells to produce their own erythrocytes, a sub-topic of the general Review topic that's underappreciated and under-discussed in other Reviews. In my opinion, they could mention some recent publications in this part of the manuscript, including (yet not limited to) 10.1016/j.semcancer.2021.10.005. Apart from the above minor comments regarding the addition of some references, I believe that the authors could also mention the role of mature erythrocytes in the tumor microenvironment. These erythrocytes enter the tumor microenvironment due to often microhemorrhages. In my opinion, despite the focus of this review on erythrocyte precursors, since the mature erythrocytes also affect cancer immunity, they should be acknowledged in such a review (e.g., 10.2174/1573394718666220428120818; 10.4049/jimmunol.1400643). Another minor comment is that the authors could restructure a bit the second Figure since it contains a lot of information and it's quite difficult to follow. Please also add a dash in anti-tumor (line 52), so that the reader can better understand that pro- also accompanies tumor, and proofread the manuscript for some minor language mistakes. Overall, I truly believe that this review contains a lot of important information and will be useful for the scientific community.

There are some minor english mistakes throughout the manuscript, but overall the writing is fine.

Author Response

In their review titled " Erythroid cells as full participants in the tumor microenvironment" Shevchenko et al discuss the role of erythroid progenitors in the tumor microenvironment. This subject has been in the spotlight of cancer research in the last years, since erythroid progenitors seem to function in an immune-modulatory way in the tumor microenvironment, helping the cancer cells survive. Therefore, the topic the authors chose is original and relevant to the field of cancer biology and cellular and molecular sciences. While there are some recent reviews on this topic (maybe the authors could acknowledge them in their revised version, e.g., 10.1016/j.canlet.2023.216193, 10.1097/MOH.0000000000000762 and 10.1515/biol-2022-0102), the authors draw a quite complete picture of cancer cases in which the role of erythroid progenitors has been explored, including melanoma, hepatic carcinoma, and Lewis carcinoma, and discuss the pleiotropic ways in which erythroid cells act immunosuppressively (graphically shown in Figure 1).

We have studied the articles you have proposed with great interest and have included interesting information from them in our review. We have marked these fragments in blue.

The authors also discuss the impressive yet monstrous ability of some cancer cells to produce their own erythrocytes, a sub-topic of the general Review topic that's underappreciated and under-discussed in other Reviews. In my opinion, they could mention some recent publications in this part of the manuscript, including (yet not limited to) 10.1016/j.semcancer.2021.10.005.

We have added some information from several interesting papers on polyploid giant cells and their differentiation into erythroid cells.  We have marked these fragments in blue.

 Apart from the above minor comments regarding the addition of some references, I believe that the authors could also mention the role of mature erythrocytes in the tumor microenvironment. These erythrocytes enter the tumor microenvironment due to often microhemorrhages. In my opinion, despite the focus of this review on erythrocyte precursors, since the mature erythrocytes also affect cancer immunity, they should be acknowledged in such a review (e.g., 10.2174/1573394718666220428120818; 10.4049/jimmunol.1400643).

I studied this article while writing a review. It contains very interesting information. I didn't include it because it described the properties of mature red blood cells in the tumor. I am very glad that you suggested me to use it in the review. We have marked these fragments in blue.

Another minor comment is that the authors could restructure a bit the second Figure since it contains a lot of information and it's quite difficult to follow.

We have divided this drawing into 2 parts. Now Figure 2 contains information about the stages of differentiation of erythroid cells and the presence of CD 45+ erythroid cells in organs and tissues. Figure 3 now shows the main mechanisms that CD 45+ erythroid cells use to maintain the tumor.

Please also add a dash in anti-tumor (line 52), so that the reader can better understand that pro- also accompanies tumor, and proofread the manuscript for some minor language mistakes.

We have made corrections. We have marked these fragments in blue.

Overall, I truly believe that this review contains a lot of important information and will be useful for the scientific community.

I thank you for evaluating our work